# SERINC5-Mediated Restriction of HIV-1 Infectivity Correlates with Resistance to Cholesterol Extraction but Not with Lipid Order of Viral Membrane

**DOI:** 10.3390/v14081636

**Published:** 2022-07-27

**Authors:** Gokul Raghunath, Yen-Cheng Chen, Mariana Marin, Hui Wu, Gregory B. Melikyan

**Affiliations:** 1Department of Pediatrics, Division of Infectious Diseases, School of Medicine, Emory University, Atlanta, GA 30322, USA; graghun@emory.edu (G.R.); ychen71@emory.edu (Y.-C.C.); mmarin@emory.edu (M.M.); hui.wu@emory.edu (H.W.); 2Children’s Healthcare of Atlanta, Atlanta, GA 30322, USA

**Keywords:** restriction factors, SERINC, HIV-1, lipid order, cholesterol, viral fusion, envelope protein stability, virus imaging, cyclodextrin

## Abstract

Serine incorporator 5 (SER5) is a protein that upon incorporation into virions inhibits HIV-1 infectivity by interfering with the ability of the Env glycoprotein to promote viral fusion. The mechanisms by which SER5 antagonizes HIV-1 fusion are not well understood. A recent study of SER5’s structure revealed a lipid-binding pocket, suggesting the ability to sequester lipids. This finding, along with the well-documented modulation of HIV-1 infectivity by viral lipids, especially cholesterol, prompted our examination of SER5′s effect on the general lipid order of the HIV-1 membrane. Pseudoviruses bearing the SER5-sensitive HXB2-Env and containing SER5 or SER2, a control protein that lacks antiviral activity, were analyzed using two distinct lipid-order probes. We show that SER5 incorporation does not noticeably affect the lipid order of pseudoviruses. Although viral cholesterol extraction reduces HIV-1 infectivity, SER5+ viruses are less sensitive to cholesterol extraction than the control samples. In contrast, the virus’ sensitivity to cholesterol oxidation was not affected by SER5 incorporation. The hydrolytic release of sphingomyelin-sequestered cholesterol had a minimal impact on the apparent resistance to cholesterol extraction. Based on these results, we propose that a subpopulation of more stable Env glycoproteins responsible for the residual infectivity of SER5+ viruses is less sensitive to the cholesterol content of the viral membrane.

## 1. Introduction

Human immunodeficiency virus (HIV-1) remains a global public safety threat. The natural infection cycle of this enveloped virus involves the fusion of the viral membrane with the host cell membrane upon entry and the envelopment of the viral capsid/core upon assembly leading to budding at the plasma membrane [1,2]. The envelope glycoproteins (Env) on the surface of HIV-1 facilitate infection by mediating the fusion between the viral and cellular membranes [3]. The trimeric Env glycoprotein complexes are made up of two subunits, the surface subunit gp120 and membrane-spanning subunit gp41 [4,5]. Only a few Env trimers incorporate into HIV-1 particles [6] and, on mature virions, these trimeric complexes have been reported to mostly form single clusters consisting of several Envs [7,8,9]. Such Env clustering has been proposed to increase their local Env density and thereby help overcome the steep energy barrier for fusion between the viral and host membranes [7,10]. However, other studies have reported random distributions of Env trimers, suggesting fewer Env trimers are needed for HIV-1 fusion (e.g., [11]).

HIV-1 assembly takes place primarily in the sphingolipid and cholesterol-rich domains of the plasma membrane, known as the lipid rafts [12,13]. Accordingly, quantitative studies have established that the phospholipid composition of HIV-1 viruses tends to be more raft-like, with a high cholesterol and sphingomyelin (SM) content relative to the lipid composition of the infected cells [14,15]. Several studies have demonstrated the importance of specific lipids and lipid domains for HIV-1 infectivity, including the role of gangliosides in virus attachment and uptake by immune cells (reviewed by Garai et al. [16]). Other studies have demonstrated the enrichment of phosphoinositides in the viruses relative to the plasma membranes of the producer cells [17,18]. Raft-forming lipids, cholesterol, and sphingomyelin are known to regulate HIV-1 fusion competence and infectivity [19]. Indeed, the disruption of such rafts using a variety of techniques has been shown to inhibit HIV-1 infectivity [20,21]. The sequestration of membrane cholesterol using amphotericin B methyl ester has been shown to inhibit HIV-1 infectivity [22]. Additionally, the truncation of the cytoplasmic tail of gp41 led to the viral evasion of inhibition by amphotericin B, further cementing the importance of membrane cholesterol and raft formation in viral infection [22]. HIV-1 particles produced by cells lacking ceramide synthase, an enzyme that is important for regulating cellular sphingolipids, have decreased infectivity without any defects in the overall virus assembly or glycoprotein incorporation [23].

Although the mechanism by which cholesterol modulates HIV-1 Env function is not well understood, there is evidence to show that the gp41 subunit interacts with cholesterol through a CRAC motif (cholesterol recognition amino acid consensus motif) located in the membrane-proximal external region (MPER) of the subunit [24,25]. Disruption to these motifs has been shown to interfere with Env function [24,26]. Other than the CRAC motif in the MPER region of gp41, recent work from Garai et al. has reported that the interactions between the gp41 C-terminal tail and cholesterol are essential for HIV-1 fusion and Env clustering [27]. By utilizing a variant of Env that carries a truncated cytoplasmic tail of gp41, they were able to show that the residues 751–854 present in the cytoplasmic tail are necessary for single Env cluster formation, cholesterol binding, and fusion competence [27].

In addition to lipid composition, the HIV-1 fusion activity is regulated by host-restriction factors, including serine incorporator 5 (SERINC5, abbreviated as SER5 henceforth) [28,29,30]. Several studies have revealed a multi-faceted restriction mechanism of SER5 through the impairment of fusion pore formation and the promotion of the functional inactivation of the HIV-1 Env [30,31,32]. We and others have shown that SER5 can induce conformational changes in Env, as evidenced by altered antibody binding and neutralization sensitivity [31,33]. Super-resolution studies on single pseudoviruses by our group have shown that SER5 incorporation can impede Env clustering on virions, thereby restricting fusion efficiency through Env rearrangement [34]. Importantly, SER5 and Env did not colocalize on single virions, indicating that the SER5-induced restriction of the Env function is indirect. The cryo-EM structure of SER5 from drosophila reveals that the protein may carry a putative cholesterol binding pocket [35]. By using a combination of cryo-EM and molecular dynamics simulation, another study has shown that the SERINC family member, SER3, can potentially bind to cholesterol and other lipids and induce lipid flipping after incorporation into the viruses [36]. Interestingly, lipidomic analysis of HIV-1 pseudoviruses incorporating SER5 shows that, despite direct and functional incorporation of SER5 into viruses, the overall lipid composition is largely unaffected [37]. These observations combined with evidence that direct gp41–cholesterol interaction plays a critical role in Env clustering and functionality [27], suggest that SER5 might inhibit HIV-1 fusion by reorganizing local cholesterol distribution.

In another study, Sundaram et al. were able to correlate the cholesterol content with the infectivity and lipid order of HIV-1 pseudoviruses [20]. It has also been shown that the enzymatic oxidation of viral lipid cholesterol can inhibit HIV-1 infectivity [21,27], presumably due to the inability of the oxidized cholesterol to form stable rafts in the viral membrane. However, another study concluded that HIV-1 resistance to cholesterol extraction correlates with the intrinsic stability of Env and not with lipid order [38]. Thus, the link between virus infectivity and lipid order remains elusive.

Here, we sought to understand the role of viral phospholipid organization and cholesterol in SER5-mediated inhibition of HIV-1 infection. We employed a combination of fluorescent lipid order probes and the manipulation of viral cholesterol to provide mechanistic insights into the SER5-mediated restriction of HIV-1 infection. By adopting a multi-pronged experimental approach involving both imaging and infectivity-based assays, we show that SER5 incorporation into HIV-1 particles does not appreciably affect their lipid order and that minute changes in lipid order were not linked to a reduction in infectivity. Interestingly, infectivity measurements reveal that SER5-containing viruses are less sensitive to cholesterol extraction, whereas the associated decreases in lipid order were similar for SER5-containing and control viruses. The resistance to cholesterol extraction in SER5-containing viruses was also observed in viruses pre-treated with sphingomyelinase (SMase). We also find that SER5 viruses do not exhibit any appreciable resistance to cholesterol oxidation or SMase treatment on their own, indicating that the apparent resistance to cholesterol disruption is unique to the extraction treatment using methyl-β-cyclodextrin (MβCD).

## 2. Materials and Methods

### 2.1. Plasmids, Cells, Lipids, and Chemicals

HeLa-derived TZM-bl cells were obtained from the NIH HIV Reagent Program, Division of AIDS, NIAID, NIH. The cells were grown in high-glucose DMEM with 10% heat-inactivated fetal bovine serum (FBS) (Atlanta Biologicals, Flowery Branch, GA, USA) and 100 units/mL penicillin/streptomycin (Gemini Bio-Products, Sacramento, CA, USA). HEK293T/17 cells were acquired from ATCC (Manassas, VA, USA) and grown in medium supplemented with 0.5 mg/mL G418 (Cellgro, Mediatech, Manassas, VA, USA).

High-glucose DMEM with and without phenol red were obtained from Corning (Glendale, AZ, USA) and Cytiva (Marlborough, MA, USA), respectively. The Lenti-X concentrator was acquired from Takara Bio (San Jose, CA, USA). The 4-(2-hydroxyethyl)-1-piperazineethanesulfonic acid (HEPES) 1 M solution was purchased from Cytiva. The Bright-Glo luciferase kit was purchased from Promega (Madison, WI, USA). Poly-L-lysine, poly-D-lysine, Methyl-β-cyclodextrin, 5(6)-Carboxyfluorescein, Cholesterol Oxidase from *Streptomyces* sp., and Sphingomyelinase from *Bacillus cereus* were purchased from Sigma (St. Louis, MO, USA).

Laurdan (6-dodecanoyl-2-dimethylaminonaphthalene; cat # D250) and Nile Red (9-(diethylamino)-5H-benzo[a]phenoxazin-5-one; cat # N1142) were obtained from Thermo Fisher Scientific (Waltham, MA, USA). Atto-647N-NHS Ester (cat # 18373) was purchased from Sigma-Aldrich. Anti-human Alexa Fluor-647 (cat # A21445) and mouse anti-HA.11 (cat # 901501) antibodies were purchased from Invitrogen (Waltham, MA, USA) and BioLegend (San Diego, CA, USA), respectively. Goat anti-mouse CF568-conjugated (cat # 20800), goat anti-human HRP-conjugated (cat # 31412), and goat anti-mouse Abberior STAR RED antibodies were acquired from Biotium (Fremont, CA, USA), Thermo Fisher Scientific, and Abberior GmbH (Göttingen, Germany), respectively. A 16% paraformaldehyde (PFA) stock (cat # 28908) was purchased from Thermo Fisher Scientific. The HIV protease inhibitor Saquinavir (cat # 4658), human 2G12 antibody (cat # 1476), and human HIV immunoglobulin (HIV IG) (cat # 3957) were obtained from the NIH HIV Reagent Program.

1,2-Dioleoyl-sn-glycero-3-phosphocholine (DOPC), Cholesterol, N-stearoyl-D-erythro-sphingosylphosphorylcholine (SM), 1,2-dioleoyl-sn-glycero-3-phosphoethanolamine-N-(7-nitro-2-1,3-benzoxadiazol-4-yl) (NBD-PE), 1,2-dioleoyl-sn-glycero-3-phosphoethanolamine-N-(Cyanine 5) (Cy5-PE), 1,2-dioleoyl-sn-glycero-3-phosphoethanolamine-N-(cap biotinyl) (biotin-PE) were purchased from Avanti Polar Lipids (Alabaster, AL, USA).

The pCAGGS plasmid encoding HIV-1 HXB2 envelope glycoprotein, the pR9ΔEnvΔNef HIV-1-based packaging vector, and pcRev plasmids have been described previously [39]. The CMV-SERINC2-iHA and CMV-SERINC5-iHA vectors were kindly provided by Dr. M. Pizzato [34]. The GFP-Vpr and mRFP-Vpr plasmids were gifts from Dr. T. Hope (Northwestern University, Evanston, IL, USA). pET28a-Lysenin-WT was a gift from Dr. Kulma (Nencki Institute of Experimental Biology, Warszawa, Poland). Wild-type Lysenin was expressed in *E. coli* BL21(DE3) based on the protocol described elsewhere [40]. Subsequently, the purified protein was labeled with Atto-647N based on the manufacturer’s labeling protocols.

### 2.2. Virus Production and Purification

The HIV-1 pseudoviruses were produced by transfecting HEK293T/17 cells with JetPRIME transfection reagent (Polyplus-transfection, Illkirch-Graffenstaden, France). To produce viruses for Laurdan staining, HEK293T/17 cells seeded in 10 cm dishes were transfected with 2.68 μg of HXB2 Env, 3.56 μg of pR9ΔEnvΔNef, 1.2 μg of pcRev, 0.3 μg of mRFP-Vpr, 0.3 μg of either CMV-SERINC2-iHA, CMV-SERINC5-iHA, or empty pcDNA3.1 vector. To generate viruses for NR staining, the same plasmids were used except that 0.6 μg of GFP-Vpr was used instead of mRFP-Vpr. The HEK293T/17 cells and DNA transfection mix were incubated overnight at 37 °C in 5% CO_2_ after which time the transfection medium was replaced with high-glucose DMEM without phenol red with 10% FBS. To produce immature viruses, DMEM containing 300 nM Saquinavir was used to inhibit HIV-1 protease. Forty-eight hours post-transfection, the supernatants were collected, passed through 0.45 µm filters, concentrated with Lenti-X concentrator, aliquoted, and stored at −80 °C. The p24 content of viral preparations was determined by ELISA, as previously described [41,42].

For infectivity assays, TZM-bl cells seeded in black clear-bottom 96-well plates (Corning) at 0.2 × 10^5^ cells/well were infected with viruses in high-glucose DMEM with 10% FBS and 20 mM HEPES by centrifugation at 4 °C for 30 min at 1550× *g*. Forty-eight hours later, infected cells were lysed and incubated with Bright-Glo luciferase substrate at room temperature, then immediately measured with a TopCount NXT reader (PerkinElmer Life Sciences, Waltham, MA, USA). The results were normalized to the p24 content.

### 2.3. Large Unilamellar Vesicles Preparation 

Large unilamellar vesicles (LUVs) were prepared using the extrusion techniques described previously [43,44]. Briefly, lipid mixtures were dissolved in chloroform and dried under a gentle stream of argon gas on the inner wall of a small glass vial and then placed under vacuum for at least two hours to form lipid cakes. For preparing liposomes with varying lipid orders, the lipid cakes were prepared using the following molar ratios of DOPC-Cholesterol-SM: (1) 1:0:0 (Ld), (2) 0.67:0.33:0, (3) 0.57:0.33:0.1, (4) 0.33:0.33:0.33, (5) 0.1:0.33:0.57, and (6) 0:0.33:0.67 (Lo), respectively. To all the lipid mixtures, 2 mol% of biotinB-PE was added to facilitate immobilization on the coverslips. Additionally, a small amount (0.1–0.3 mol%) of either NBD-PE or Cy5-PE was added to act as fiducial markers for the NR and Laurdan imaging, respectively.

The mixtures were hydrated using 1 mL of PBS−/− and vortexed thoroughly to create multilamellar structures, followed by 5 freeze-thaw cycles. The resulting solutions were extruded through a polycarbonate membrane (pore size 100 nm; Whatman/GE Healthcare) 20 times utilizing a mini extruder (Avanti Polar Lipids, Alabaster, AL, USA) to produce a clear solution of 100 nm unilamellar liposomes. For soluble dye-loaded liposomes, the mixtures were first hydrated with a 1 mL solution of 1 μM 5,6 carboxyfluorescein and then subjected to extrusion. After extrusion, the liposomes were subjected to dialysis to remove the excess dye.

For immobilizing intact liposomes, we adopted a strategy previously described [45]. Briefly, a coverglass surface was coated with BSA and biotin-conjugated BSA at a 100:1 molar ratio. After washing the surface thoroughly with PBS−/−, the surface was treated with 0.01–0.025 mg/mL streptavidin solution for 30 min. This creates a uniform layer for the biotinylated liposomes to bind. Finally, a 1–4 μM suspension of liposomes was added to immobilize the liposomes on the surface.

### 2.4. Lipid Probe Staining

A Laurdan stock solution, 1 mg/mL, was prepared in DMSO at 50 °C. After the Laurdan crystals were fully dissolved, the Laurdan stock solution was aliquoted, purged with Argon, and stored at −20 °C. Each aliquot was used within a month and purged with Argon right after the solution contacted the ambient air. For Laurdan staining, 0.1 mM liposomes or viruses (1–2 ng p24) were mixed with 28 µM Laurdan while vortexing. The viruses and Laurdan mixtures were incubated at 37 °C for 10 min, diluted 50-fold, and attached to poly-D-lysine (PDL)-coated coverglass for single virus imaging.

NR stock was prepared by dissolving 1 mg of the dye in 1 mL DMSO at room temperature. Once fully dissolved, the stock solution was purged with Argon and stored at −20 °C. Each aliquot was used within 3 months of preparation. First, 1–2 ng of virus p24 diluted in PBS+/+ were attached to a PDL-coated coverglass. Then, the samples were blocked by 0.1 mg/mL BSA for 30 min and the surface was washed with PBS+/+. Immediately prior to imaging, a 10 μM solution of NR was prepared in PBS+/+ and diluted to 500 nM in the imaging chamber for image acquisition. The solution was prepared immediately before the experiment and used within an hour of preparation to prevent artifacts in imaging due to dye aggregation over time.

For measuring the lipid order changes after lipid manipulation treatments, we followed a similar protocol to the one mentioned above. Laurdan-stained viruses were immobilized and treated with either MβCD, COase, SMase, or SMase-MβCD at concentrations specified below for 30 min at 37 °C. Post-treatment, the samples were brought to RT and washed thoroughly with PBS+/+ prior to imaging. For NR imaging, the surface-immobilized viruses were first blocked using 0.1 mg/mL BSA for 30 min and washed thoroughly with PBS+/+. Then, the samples were subjected to a lipid manipulation treatment for 30 min at 37 °C. After thoroughly washing the treated samples, a diluted NR solution was added to a final concentration of 500 nM in the imaging chamber for acquisition.

### 2.5. Infectivity Measurements of Pseudoviruses Subjected to Lipid Manipulation Treatments

For accurately testing the effects of manipulating viral lipids (MβCD, COase, SMase, and SMase-MβCD) on the infectivity of pseudovirus samples, we used the Luciferase-based infectivity assay (see Section 2.2) with minor modifications to the protocol. We first attached 0.02–0.2 ng p24 of the viruses to a PDL-coated 96-well plate for 30 min at 4 °C. The p24 content of the viruses was adjusted to produce a robust Luciferase signal-to-noise ratio, especially for the low signals of the SER5-containing viruses (Appendix A). After washing thoroughly, the immobilized viruses were subjected to treatment with MβCD/COase/SMase at the final reaction concentrations specified below. The wells were subsequently washed with PBS+/+. TZM-bl cells at 0.5 × 10^5^ cells/well in high-glucose DMEM with 20 mM HEPES and without FBS were centrifuged at 4 °C for 30 min at 1550× *g* to overlay the immobilized and treated viruses. We reasoned that the lipids present in FBS can potentially affect the lipid composition of the surface-bound viruses, as cholesterol-depleted viruses are more susceptible to exogenous lipid incorporation [21]. After incubation of the TZM-bl cells in high-glucose DMEM with 20 mM HEPES and without FBS for 2 h at 37 °C, we then added FBS to the wells to reach a final concentration of 10% FBS. After incubating the cells for 36–48 h, luminescence measurements were performed to evaluate the infectivity of the surface-bound virions.

The MβCD concentrations for the surface treatment of viruses varied from 0 mM (untreated) to 1 mM. For the COase treatments, we used from 0 U/well (untreated) to 0.1 U/well. For the SMase treatment, we used a concentration range between 0 U/well (untreated) and 0.05 U/well. All reactions were carried out for 30 min at 37 °C.

For the infectivity measurements of viruses treated with MβCD in solution, 0.05 ng p24 of viruses were incubated with 0 mM (untreated) to 1 mM MβCD in solution for 30 min at 37 °C. After the MβCD treatment, viruses were immobilized on a PDL-coated 96-well plate for 30 min at 4 °C. After washing thoroughly with PBS+/+, we overlayed the TZM-bl cells in the same way as in the infectivity assays of the surface-treated viruses. After a 2 h incubation with no FBS, 10% FBS was added and 36–48 h later, the infected cells were lysed and incubated with a Bright-Glo luciferase substrate at room temperature and immediately measured with a TopCount NXT reader.

For subsequent treatments of SMase and MβCD, we first treated the immobilized viruses at 0.05 U/well SMase at 37 °C for 30 min. We then washed the surface thoroughly with PBS+/+ and subjected the samples to an additional 30-minute treatment at 37 °C, with the [MβCD] as specified above. After the incubation was complete, the 96-well plate was thoroughly washed, and we then overlaid the TZM-bl cells atop the bound pseudovirus samples as previously mentioned. An amount of 10% FBS was added after the 2 h incubation with no FBS and 36–48 h later, the infected cells were lysed and incubated with a Bright-Glo luciferase substrate at room temperature and immediately measured with a TopCount NXT reader.

### 2.6. Lipid Order Imaging and Analysis

Single virus imaging was performed on a Zeiss LSM880 (Carl Zeiss Microimaging, Jena, Germany) with a 40× water objective. Laurdan was excited using a 405 nm laser and the emission was collected in Lambda mode between 411 nm and 553 nm in 8.9 nm wide intervals (16 intervals total) [46]. The Laurdan GP values were calculated using the coordinates of the positive mRFP-Vpr signals for virus imaging and the Cy5-PE signals for liposome imaging:(1)$$Laurdan\;GP=\frac{I_{443\mathrm{nm}}-I_{513\mathrm{nm}}}{I_{443\mathrm{nm}}+I_{513\mathrm{nm}}}$$

NR was excited using a 561 nm laser and the emission data was by using the emission windows between 562 and 606 nm and 607 and 651 nm, respectively. The NR GP values were calculated using the following equation on puncta exhibiting coordinates of the positive GFP-Vpr signals for virus imaging and the NBD-PE signals for liposome imaging:(2)$$Nile\;Red\;GP=\frac{I_{\left(562\mathrm{nm}-606\mathrm{nm}\right)}-I_{\left(607\mathrm{nm}-651\mathrm{nm}\right)}}{I_{\left(562\mathrm{nm}-606\mathrm{nm}\right)}+I_{\left(607\mathrm{nm}-651\mathrm{nm}\right)}}$$

The signals of the virus (GFP-Vpr, mRFP-Vpr) and liposomes (NBD-PE, Cy5-PE) were detected using a wavelet-based localization algorithm by ICY software [47]. Binary masks were created from ROIs of localized viruses or liposomes with 3-pixel dilation. Masked Laurdan or NR signals were localized again using a wavelet-based algorithm with a proper threshold to avoid a low signal-to-background ratio and big dye aggregates. All Laurdan and NR signals were background subtracted and used for subsequent GP value calculations.

### 2.7. Immunofluorescence Imaging and Analysis

SERINCs-iHA-containing viruses were immobilized to PDL- or PLL-coated coverslips for 30 min at room temperature, fixed by 2% PFA for 30 min at room temperature, and then blocked by PBS++ with 15% FBS. The samples were incubated with anti-HA antibody, followed by staining with anti-mouse AF488 or anti-mouse antibodies conjugated with STARRED or CF568 for the mRFP-Vpr (Laurdan imaging) or GFP-Vpr (Nile Red imaging) pseudoviruses, respectively. The immunofluorescence signals of SERINCs-iHA were acquired using an Elite DeltaVision microscope (Cytiva, Marlborough, MA, USA) or Zeiss LSM880.

The GFP-Vpr (Nile Red imaging) and mRFP-Vpr (Laurdan imaging) signals were first localized using the local maximum adapted from fastpeakfind by Adi Natan (https://www.mathworks.com/matlabcentral/fileexchange/37388-fast-2d-peak-finder (accessed on 14 September 2018)). After discarding the top 15% virus signals to remove possible virus aggregates and a signal-to-background ratio smaller than 1.5 to remove the dim virus signals, the coordinates of the Vpr spots were used to acquire the background-subtracted fluorescence signals of SERINCs-iHA immunostaining. The medians of SER2 and SER5 immunofluorescence were acquired for calculating the SER2/SER5 ratio.

### 2.8. Statistical Analysis

Differences in lipid order on untreated pseudoviruses were compared using the Kolmogorov–Smirnov test. The lipid orders of different viruses treated with multiple concentrations of raft-perturbing reagents (MβCD, Coase, and Smase) were compared with 2-way ANOVA repeated measures. The infectivity measurements of different viruses treated with multiple concentrations of raft-perturbing reagents were fitted by Hill–Langmuir equation with variable slope,
(3)$$Y=\frac{100}{\left(1+{\left(\frac{IC50}{X}\right)}^{a}\right)}$$
where *X* is the concentration of MβCD, COase, or SMase, and *Y* is the normalized infectivity ratio (100%) by untreated viruses. *IC*50 and *a* were the fitting parameters. The same datasets were compared by 2-way ANOVA repeated measures for statistical significance. All the statistical analyses and curve fitting were conducted with GraphPad Prism 9.3.1 (San Diego, CA, USA).

## 3. Results

To investigate the effects of SER5 on HIV-1 lipid order, we used two fluorescent probes, Nile Red and Laurdan. Laurdan is a popular reporter of lipid order that does not preferentially partition into specific lipid domains [48,49,50]. This lipophilic dye senses the polarity of its lipid–water interface by the fluorescence moiety, exhibiting a red shift in the fluorescence emission when incorporated in liquid-disordered regions of the lipid bilayer. Nile Red (abbreviated henceforth as NR) is a lesser known environmentally sensitive probe that also reports lipid order [51,52,53]. NR is a solvatochromic dye with a fluorescence emission that is strongly dependent on the polarity of the neighboring solvent. NR is known to exhibit a non-emissive twisted intramolecular charge transfer state that is better stabilized in relatively polar environments, leading to a red shifting of the fluorescence emission of the dye when present in liquid-disordered regions of the lipid bilayer [54]. The solvatochromicity of NR has enabled quantitative measurements of the lipid phase structures of cell membranes, cell-derived structures, and liposome formulations [52,55,56]. However, quantitative measurements of the lipid order in the context of viruses remain lacking.

### 3.1. Laurdan and Nile Red Are Reliable Probes for Measuring Lipid Order of Large Unilamellar Vesicles

To determine the utility of NR as a lipid order probe for pseudovirus particles, we first tested the dye on liposomes made with varying lipid compositions. We chose large unilamellar liposomes as our model system for the calibration of lipid order reported by NR, owing to their controllable lipid compositions and sizes (~100 nm) that closely resemble the diffraction-limited size of individual pseudovirus particles. We tested a wide range of liposomes made with ternary compositions of varying lipid orders by modifying the amount of sphingomyelin (SM) incorporated during the liposome extrusion process. First, we measured the NR fluorescence by incubating the dye in a solution with the liposomes using a 96-well plate reader (Figure 1A). The NR fluorescence shifted from a ~580 nm peak for lipid-ordered liposomes (Lo) to a ~630 nm peak for liquid-disordered liposomes (Ld), indicating that the solvatochromic nature of NR can accurately report on the lipid order of the liposomes tested in our experiments. Similarly, the fluorescence spectra of Laurdan mixed with liposomes shifted from a ~440 nm peak for Lo to a ~510 nm peak for Ld (Figure 1B), and a consistent shoulder peak at ~440 nm was likely from the Laurdan aggregates (data not shown).

To quantitatively assess the shift in the fluorescence emission, we utilized Generalized Polarization (GP), a ratiometric function (see Methods) that uses spectral data to provide information on the lipid order of the samples tested (Figure 1C). Our measurements using a 96-well plate reader showed a clear shift in the GP values ranging from ~−0.22 for Ld to ~0.21 for Lo liposomes, in line with the established literature that uses NR GP [52]. These data imply that NR GP can be used as a reliable metric for lipid-order measurements. To verify the validity of the data reported using NR, we performed similar experiments using the dye Laurdan. Laurdan GP (see Methods) values ranged from ~−0.2 for Ld to ~0.4 for Lo liposomes (Figure 1D).

Although the solution measurements provide a simple high-throughput method of measuring the lipid order of liposomes, achieving such measurements on pseudoviral particles is challenging owing to a limiting sample concentration. To mitigate this problem, we turned to the imaging of NR-stained surface-immobilized liposomes (see Methods). As expected, the GP values shift from ~−0.21 to ~0.06 for immobilized Ld and Lo liposomes, respectively. Although the dynamic range of GP values appears to be smaller than for bulk measurements, possibly due to NR nonspecifically binding to the surface (Appendix A), NR is still a reliable probe as evidenced by a clear correlation between the lipid order and GP values (Figure 1E). To mirror our imaging experiments with NR, the same liposome formulations were stained with Laurdan (see Methods). Owing to its low solubility in aqueous solution, Laurdan forms fluorescent aggregates that can complicate the interpretation of fluorescence data. To address that, we used the localized coordinates from Cy5-lipid-labeled liposomes to differentiate the Laurdan signals of liposomes from non-specific Laurdan aggregates (Appendix A). The Laurdan GP measured using single liposome fluorescence imaging (Figure 1F) was consistent with the Laurdan GP measured by the plate reader. The calibration of GP values on liposomes of varying compositions using Laurdan corroborates our insights from the NR GP data, indicating that NR is a reliable probe for measuring lipid order through ratiometric imaging (Figure 1E,F).

### 3.2. Lipophilic Probes Reveal a Highly Ordered Phase Structure in HIV-1 Pseudovirus Membranes

Having calibrated the GP values for liposomes, we then sought to measure the lipid order of HIV-1 pseudoviruses immobilized on poly-lysine-coated surfaces. First, we tested our dyes on two different pseudovirus preparations labeled with GFP-Vpr (for NR measurements) or mRFP-Vpr (for Laurdan measurements). The first preparation was to control viruses produced with the HXB2 HIV-1 Env glycoprotein (abbreviated as control). To test the role of HIV-1 maturation on the lipid order of the viruses, we also produced control viruses with Saquinavir (abbreviated as SQV) supplemented to the growth medium during and post-transfection of the virus-producing cells. Saquinavir is an HIV-1 protease inhibitor that prevents the proteolytic processing of Gag polyprotein leading to the production of immature pseudovirus particles that are incapable of infection [7].

By employing the image analysis method used for liposome calibration, we compared the GP values obtained from imaging pseudoviruses over multiple biological replicates. The GFP-Vpr and mRFP-Vpr fluorescences were used to distinguish the NR and Laurdan signals from the viruses bound to the surface from background fluorescence. The NR and Laurdan GP values of HIV-1 pseudoviruses were found to be ~0.06 (Figure 2A) and ~0.4 (Figure 2B), respectively. In comparison to the calibration data from the liposomes, these high GP values indicate a highly ordered lipid phase structure in viruses that is consistent with previously published studies that measured Laurdan GP values on similar pseudovirus samples [20,57]. In comparison with the control viruses, both NR and Laurdan imaging indicates a more disordered GP value for the SQV samples, which is consistent with previously published work [46].

### 3.3. SER5 Incorporation Has Minimal Impact on the Global Lipid Order of the Viral Membrane

Having established both Laurdan and NR to be reliable indicators for the viral lipid order, we sought to test if SER5 incorporation into the pseudoviruses, which strongly restricts HXB2 HIV-1 infectivity [33], can lead to appreciable differences in the lipid order. Since the incorporation of SER2, a member of the SERINC family of proteins, does not inhibit infectivity (Appendix A), we used SER2 as a control to link changes in the global lipid order (if any) to the reduction in infectivity [30]. We report minor decreases in the GP values between the control and SER5-containing or SER2-containing viruses without a consistent trend with regard to statistical significance from both the Laurdan and NR imaging (Figure 2). For instance, SER2 viruses were found to exhibit lower overall GP values in comparison with the control viruses in a statistically significant manner regardless of the probe used (Figure 2C,D). However, SER5 exhibits a statistically different decrease in Laurdan GP (Figure 2D) but not in NR GP (Figure 2C). Importantly, we observe insignificant differences in lipid order between SER5- and SER2-incorporated virions (Figure 2C,D). Taken together, both the NR and Laurdan data indicate that both SER5 and SER2 incorporation mildly decreases the global lipid order of the pseudoviruses. We cannot rule out the possibility that a large GP variance may mask the minute differences between the SER5 and SER2 effects on the lipid order.

### 3.4. SER5 Incorporation Renders Pseudovirus Infectivity Less Sensitive to Cholesterol Extraction by MβCD

We next extended our investigation to understand the role of cholesterol in regulating the SER5-mediated restriction of HIV-1 infection. To accomplish this, we evaluated the effects of depleting viral membrane cholesterol using MβCD. MβCD is commonly used for extracting membrane cholesterol, and its mechanisms have been studied extensively [58,59]. Given that membrane cholesterol can simultaneously affect the lipid order of the viruses and regulate infectivity [20,38], we measured both the infectivity and lipid order changes as a function of the MβCD concentration [MβCD]. The control, SER5, and SER2 samples were immobilized on the surface and subjected to cholesterol depletion with a range of [MβCD] for 30 min at 37 °C.

NR and Laurdan imaging reveal a decrease in overall pseudovirus GP as a function of the MβCD concentration (Figure 3A,B). Interestingly, no major difference was observed with regard to the lipid order changes relative to the untreated samples (measured by ∆GP, Figure 3C,D) across [MβCD], regardless of SER5/2 incorporation. This indicates that SER5 incorporation does not affect susceptibility to global membrane order reduction by cholesterol depletion.

To study the effects of cholesterol depletion on the infectivity of pseudoviruses, we performed a modified version of a luciferase-based infectivity assay established previously [60] (for details, see Methods). Unsurprisingly, a concentration-dependent reduction in overall infectivity was observed as a function of [MβCD] in both the control and SER2 samples with an apparent IC_50_ of ~0.2 mM MβCD (Figure 3F). Strikingly, SER5-incorporated viruses require a much higher [MβCD] to exhibit a comparable decrease in infectivity (IC_50_~0.4 mM), demonstrating that SER5-incorporated pseudoviruses are less sensitive to cholesterol extraction using MβCD in comparison to the control and SER2 samples. To ensure that this apparent resistance to cholesterol extraction in SER5 viruses is not a result of artifacts from the varying SER expression and virus incorporation levels, we performed immunofluorescence measurements of several of our independent viral preparations (Appendix A). We observe the consistent incorporation of SER5 and SER2 in both the GFP-Vpr and mRFP-Vpr viruses, which is consistent with previous reports [30,34]. Notably, SER2 incorporated into pseudoviruses somewhat better than SER5, implying that the observed resistance of SER5-containing pseudoviruses to cholesterol extraction is not a result of the different levels of these proteins in our pseudovirus preparations.

In addition to the MβCD treatment of surface-immobilized pseudoviruses, we treated viruses with MβCD in solution and confirmed that the MβCD resistance of SER5-incorporated viruses was significantly different from the control and SER2 samples (Appendix A). Given that SER5 incorporation significantly lowers pseudovirus infectivity, care must be taken to ensure that the measured luminescence intensity of the samples falls within the dynamic range of the assay. To that end, infectivity measurements were performed to ensure that the signal intensity from SER5 viruses was consistently maintained above the background signal levels (Appendix A). By comparing both the normalized infectivity data (Figure 3F) and non-normalized infectivity data (Appendix A), we are confident that the statistically significant changes to the apparent IC_50_ values from the MβCD treatment are a clear indication of the phenotypic differences in sensitivity to cholesterol extraction in SER5 viruses relative to the SER2 and control viruses.

### 3.5. SER5 Incorporation Does Not Affect Pseudovirus Sensitivity to Cholesterol Oxidation

With novel insights from the cholesterol depletion measurements, we then asked if SER5-containing viruses would exhibit a similar resistance to cholesterol oxidation. Unlike MβCD, cholesterol oxidase (COase) catalyzes the oxidation of cholesterol to 4-cholesten-3-one, a lipid known to have deleterious effects on the raft formation in plasma membranes and to reduce HIV-1 infectivity [61,62]. We explored the effects of varying [Coase] on control, SER5, and SER2 pseudoviruses using a combination of infectivity measurements and NR staining. As expected, the overall GP values of the viruses decrease as a function of [Coase] indicating that the oxidation of the viral membrane cholesterol to the 4-cholesten-3-one leads to a decrease in the lipid order (Figure 4A). However, no clear pattern emerged with regard to the ∆GP of the SER5 and SER2 with varying [COase] (Figure 4B). The control samples seemingly appear less sensitive to COase treatment, but the differences between the control, SER5, and SER2 across three independent measurements were statistically insignificant. This indicates that SER5 incorporation does not modulate the effects of cholesterol oxidation on the lipid order. We could not collect Laurdan data for the COase-treated pseudoviruses owing to artifacts from the fluorescence measurements unique to the samples treated with the enzyme (Appendix A).

In stark contrast to our infectivity data on cholesterol depletion, COase treatment across a wide concentration range shows no appreciable difference between its effects on control, SER5, and SER2 infectivity (Figure 4C). This indicates that the apparent resistance to cholesterol manipulation in SER5-incorporated pseudoviruses is exclusive to the MβCD-mediated depletion of viral membrane cholesterol and not enzymatic oxidation by COase, though both MBCD and COase treatments decrease the lipid order of viral membranes.

### 3.6. SER5 Incorporation Does Not Affect Pseudovirus Sensitivity to Sphingomyelinase Treatment

The viral membrane contains high levels of cholesterol and sphingomyelin (SM), known to coexist as lipid rafts in the context of a cellular membrane. Several studies have revealed that the solvent/binding accessibility of cholesterol can be severely restricted due to its tight interaction with SM [63]. These studies have identified the existence of multiple pools of cholesterol present within the plasma membrane: accessible, inaccessible (SM-sequestered), and essential [64]. To explore the possibility of SM-induced changes in cholesterol distribution between the control, SER5, and SER2 viruses, we sought to perturb the viral membranes using SMase. SMase is an enzyme that hydrolyzes SM to ceramide, thus exposing SM-sequestered cholesterol and rearranging raft microdomains in the lipid membranes [65]. Here we performed lipid order measurements of SMase-treated viruses with both Laurdan and NR. The successful hydrolysis of viral SM upon SMase treatment was confirmed by utilizing Lysenin, a well-characterized marker for SM in the lipid membranes [66] (Appendix A). Interestingly, we observed seemingly inconsistent GP changes (ΔGP within ±0.02) in the NR-stained samples (Figure 5A,B), whereas a consistent GP decrease (ΔGP~−0.04) across SMase concentrations was seen for the Laurdan-stained viruses (Figure 5C,D). We attribute the relatively subtle changes in NR imaging to the lower dynamic range of NR in our experimental setup.

In comparison to COase and MβCD, SMase treatment appeared to have a relatively subtle impact on pseudovirus infectivity across the concentration range tested (Figure 5E). Unlike MβCD or COase, SMase-treated viruses seemed to suffer from a large variability from sample to sample, as evidenced by the error bars provided (Figure 5E). The source of this variability is currently unknown, but a roughly 50% reduction in infectivity at 0.05 U SMase treatment is comparable to similar experiments performed on Influenza A virus particles (treated with 0.1 U SMase) from a previously published study [67]. Interestingly, across multiple biological replicates, we observe no appreciable differences between the control, SER5, and SER2 samples (Figure 5E). These results indicate that the resistance to cholesterol depletion by SER5 incorporation does not extend to SM depletion using SMase treatment.

### 3.7. SER5-Containing Viruses Retain Resistance to Cholesterol Extraction despite Sphingomyelinase Treatment

To assess if altering cholesterol accessibility would impact cholesterol extraction using MβCD on viruses, we treated the viruses with SMase first, followed by MβCD at varying concentrations (for details, see Methods). It is well known that MβCD can successfully extract from both accessible and non-accessible pools of cholesterol from the plasma membrane [68]. Given that SMase treatment is known to increase accessibility to the cholesterol in membrane rafts, we wondered if our pseudovirus samples would exhibit greater sensitivity to cholesterol extraction after pre-treatment with SMase.

NR imaging shows that, following SMase treatment, MβCD treatment exhibits larger changes to the observed GP values compared to the samples not pretreated with SMase, even at low [MβCD] concentrations (Appendix A and Figure 3A). This observation indicates that improving accessibility to cholesterol with SMase treatment renders the viruses more susceptible to cholesterol depletion, culminating in a larger change in the lipid order relative to SMase-untreated viruses. Interestingly, despite the increased sensitivity to cholesterol extraction, no appreciable differences were observed between the control, SER5, and SER2 with regard to the lipid order changes (Appendix A). It is worth noting that MβCD treatment at higher concentrations (0.25 mM) reduced the number of observable NR puncta during imaging, evidenced by the decreasing fluorescence signals (Appendix A). The Laurdan-stained viruses exhibited even lower fluorescence signals at higher [MβCD] and hence were not used for subsequent analysis (data not shown). Although the reason for the decrease in the observable NR and Laurdan puncta is not clear, we speculate that due to very efficient cholesterol depletion in SMase-treated viruses, the association of lipid dyes to the viral lipids can be hindered under such conditions.

The infectivity measurements of the SMase pre-treated viruses reveal that sphingomyelin hydrolysis renders the viruses more sensitive to cholesterol extraction across the range of [MβCD] tested. We observe a ~60% reduction in the apparent IC_50_ (~0.08 mM; Figure 6) from the MβCD treatment in SMase-pretreated control and SER2 viruses relative to SMase-untreated control and SER2 viruses (~0.2 mM; Figure 3F). Interestingly, SER5-incorporated viruses require a higher [MβCD] to elicit a comparable decrease in infectivity even after SMase pre-treatment (IC_50_~0.15 mM; Figure 6), relative to the control and SER2 samples that were subjected to the same pre-treatment. (Note that due to the large errors in calculated IC_50_ values for control viruses, the difference with SER5 viruses did not reach statistical significance). These results show that the apparent resistance of infectivity to cholesterol extraction in SER5 viruses is not dependent on cholesterol accessibility due to SM sequestration. This observation is consistent with previous studies that have posited that MβCD depletes cholesterol from both the accessible and inaccessible pools even in the context of the viral lipid envelope.

## 4. Discussion

Here, we asked whether SER5 exerts its antiviral activity by modulating the global lipid order of the HIV-1 membrane. Toward this goal, we established an imaging-based assay to quantitatively measure the lipid order of pseudoviruses by utilizing two polarity-sensitive dyes, NR and Laurdan. Both dyes reliably reported changes in the pseudovirus lipid order upon the incorporation of host factors or treatment altering the virus’ cholesterol content. These experiments validate the use of NR for lipid order measurements, especially when the probe’s water solubility is important [69]. Although the results obtained with these probes were quite consistent, some differences exist. Most notably, the Laurdan GP measurements of the Ld/Lo liposomes provide a wider dynamic range compared to the NR GP values. This difference is likely due to a couple of factors. First, Laurdan emission spectra exhibit threefold higher amplitude changes in the emission window between the Lo and Ld liposomes relative to the NR signal that exhibits very limited changes in amplitude (data not shown). Second, owing to the background signal from the spurious NR puncta during our imaging experiments, we notice that the dynamic range of the NR-based GP measurements is further reduced (Figure 1C,D). Despite such minor caveats, NR and Laurdan are reliable indicators of lipid order with the ability to distinguish subtle lipid order changes that arise from changing lipid compositions (Figure 1).

We found that pseudoviruses exhibit a highly ordered lipid phase structure (Figure 2A,B) in agreement with previously published lipidomic data on HIV-1 and lipid order measurements using Laurdan [57]. Also consistent with the previous report [46], immature pseudoviruses have fewer ordered membranes (lower GP values) compared to control viruses (Figure 2C,D). Interestingly, Laurdan imaging shows a subtle yet statistically significant decrease in the lipid order for the SER2 and SER5 samples relative to the control. However, neither the Laurdan GP values nor the NR GP values for the SER2 and SER5 samples were statistically different. Since changes in the lipid order of SER5 and SER2 viruses do not correlate with the reduction in HIV-1 infectivity, we concluded that the minute SER5 effect on the global lipid order is unlikely to contribute to the restriction phenotype. The minimal impact of SER5 on the global lipid order of viruses is consistent with the unchanged lipid composition of SER5-containing HIV-1 [37].

If not through altering the global lipid order, how can SER5 inhibit HIV-1′s fusion competence? To address this central question, we turned to the viral lipid manipulation by MβCD, Coase, and SMase, respectively. Our cholesterol extraction experiments using MβCD confirm a dose-dependent reduction in the lipid order and a concomitant loss of infectivity (Figure 3). Perhaps the most unexpected finding of this study is the decreased sensitivity of SER5 viruses to cholesterol extraction compared to the control and SER2 samples (Figure 3E,F). Interestingly, SER5 did not impart any additional resistance to cholesterol oxidation in both the infectivity and lipid order measurements (Figure 4), in stark contrast to our MβCD infectivity experiments. Cholesterol oxidation by COase leads to the formation of cholestenone. The presence of cholestenone has been found to have deleterious effects on lipid domain formation, raft stability, and viral infectivity [21,27,70]. In addition, cholestenone has been reported to cause disruption to Env clustering, though the exact mechanism is unknown [27]. We believe that cholestenone formation using COase treatment in viruses can modulate Env’s ability to drive fusion and infection, either through direct interactions with the protein complex or by actively disrupting the lipid domain structure within the virus. Therefore, our data indicate that the apparent resistance to cholesterol manipulation in SER5 viruses is unique to extraction using MβCD. Surprisingly, improving cholesterol accessibility to MβCD by pretreating the viruses with SMase does not seem to affect the SER5 resistance to cholesterol depletion using MβCD alone, as evidenced by the ~2-fold higher IC_50_ for the SER5 viruses relative to the control and SER2 viruses (Figure 6). Taken together, these results imply a difference in sensitivity to cholesterol extraction in the SER5-incorporated samples.

Our results can be rationalized by considering recent studies that have explored the relative HIV-1 Env stability. Salimi et al. have shown that different strains of HIV-1 exhibit different sensitivities to cholesterol depletion [38]. Subsequent antibody neutralization and mutagenesis experiments allowed them to conclude that Env stability and sensitivity to cholesterol depletion are strongly related to one another. Additionally, it has been shown that HIV-1 Env can adopt a multitude of conformations with differing sensitivity to antibody neutralization [71,72,73]. Our group has also shown that HIV-1 Envs sensitive to SER5 restriction tend to exhibit faster spontaneous inactivation kinetics in the presence of SER5 relative to resistant Envs [30,74]. These studies indicate that SER5 preferentially inactivates a subpopulation of HIV-1 Envs with lower overall stability. The sensitivities of each Env conformation to SER5 restriction are currently unknown but published studies [75] and our own unpublished results suggest that CD4 binding sensitizes Env to SER5 restriction.

Based on these studies, we hypothesize that SER5 preferentially inactivates a subpopulation of Envs that exhibits lower overall stability (Figure 7). Given that stability and sensitivity to cholesterol extraction are correlated [38], the extraction-resistant fraction of Env on viruses can escape restriction by SER5 and is capable of host–cell fusion and subsequent infection. Indeed, in most infectivity measurements, SER5 viruses always exhibit ~5–10% of infectivity relative to control viruses, presumably due to the presence of an unrestricted fraction of the Env within the viruses (Appendix A). It is thus possible that owing to the presumed stability of the Env subpopulation that remains functionally active on SER5-containing viruses, these viruses are relatively resistant to cholesterol extraction, exhibiting a 2× higher IC_50_ for cholesterol extraction relative to WT and SER2 viruses (Figure 3F). We also considered an alternative model based on direct interaction with SER5 and cholesterol. A recent study showed that SER5 possesses a putative cholesterol-binding pocket [35]. It is therefore possible that SER5 directly associates with viral cholesterol and prevents extraction using MβCD. However, given that the amount of cholesterol molecules (~140,000 [18]) per virus far exceeds the number of SER5 incorporated in the virus [34], it appears unlikely that SER5-sequestered cholesterol can modulate the fusion activity by reducing the efficacy of cholesterol extraction. We believe that our results add novel insights to the mechanistic picture of the SER5-mediated restriction of HIV-1 infection and pave the way for further experiments that explore the interplay between the lipid order, Env–lipid interactions, and their role in regulating viral fusion and infection.

## Figures and Tables

**Figure 1 viruses-14-01636-f001:**
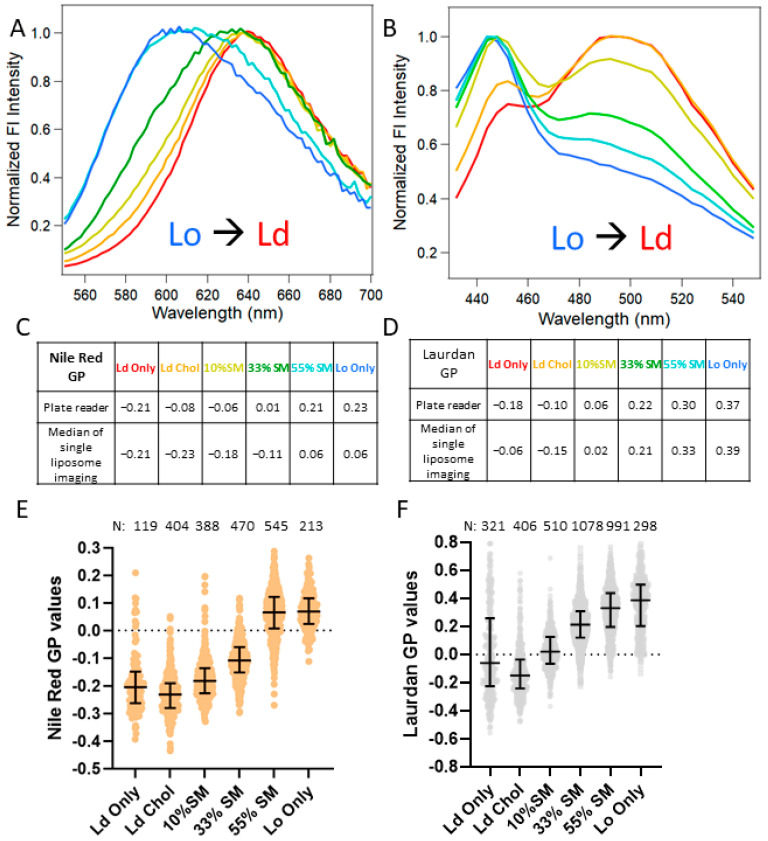
Lipid-order measurements for liposomes of different compositions using Nile Red (**A**,**C**,**E**) or Laurdan (**B**,**D**,**F**). Lipid order was measured in solution by a plate reader and presented as fluorescence spectra (**A**,**B**) or GP values (**C**,**D**) (for GP calculations, see Methods). Lo stands for liquid-ordered, and Ld stands for liquid-disordered liposomes, respectively. (**E**,**F**) Lipid order (GP) of single liposomes attached to coverglass was measured using a confocal microscope. The N representing the number of liposomes analyzed for each composition is shown above the graphs.

**Figure 2 viruses-14-01636-f002:**
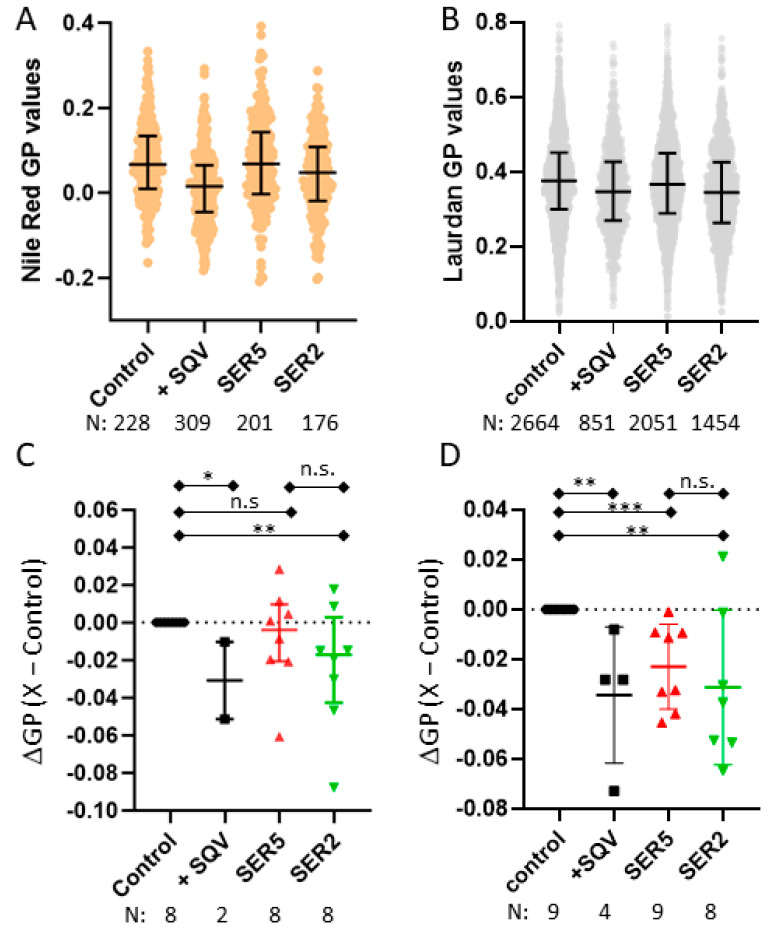
Effects of HIV-1 maturation and SERINC incorporation on the lipid order of the viral membrane. Surface-attached pseudoviruses containing either GFP-Vpr (for Nile Red imaging) or mRFP-Vpr (for Laurdan imaging) were stained with Nile Red (**A**,**C**) or Laurdan (**B**,**D**). Examples of Nile Red (**A**) and Laurdan (**B**) staining analysis on attached viruses. Horizontal black lines on the scatterplot represent 1st, 2nd, and 3rd quantiles. N, the number of viruses analyzed per experiment, is shown below the graphs. (**C**) Nile Red and (**D**) Laurdan GP changes (ΔGP) between control and +SQV-, SER5-, and SER2-incorporated pseudoviruses, where N is the number of independent measurements. ΔGP was calculated by subtracting the median of the control from the median of the +SQV-, SER5-, and SER2-containing pseudoviruses, ΔGP = X − control. n.s., *p* > 0.05; *, 0.05 > *p* > 0.01; **, 0.01 > *p* > 0.001. ***, *p* < 0.001.

**Figure 3 viruses-14-01636-f003:**
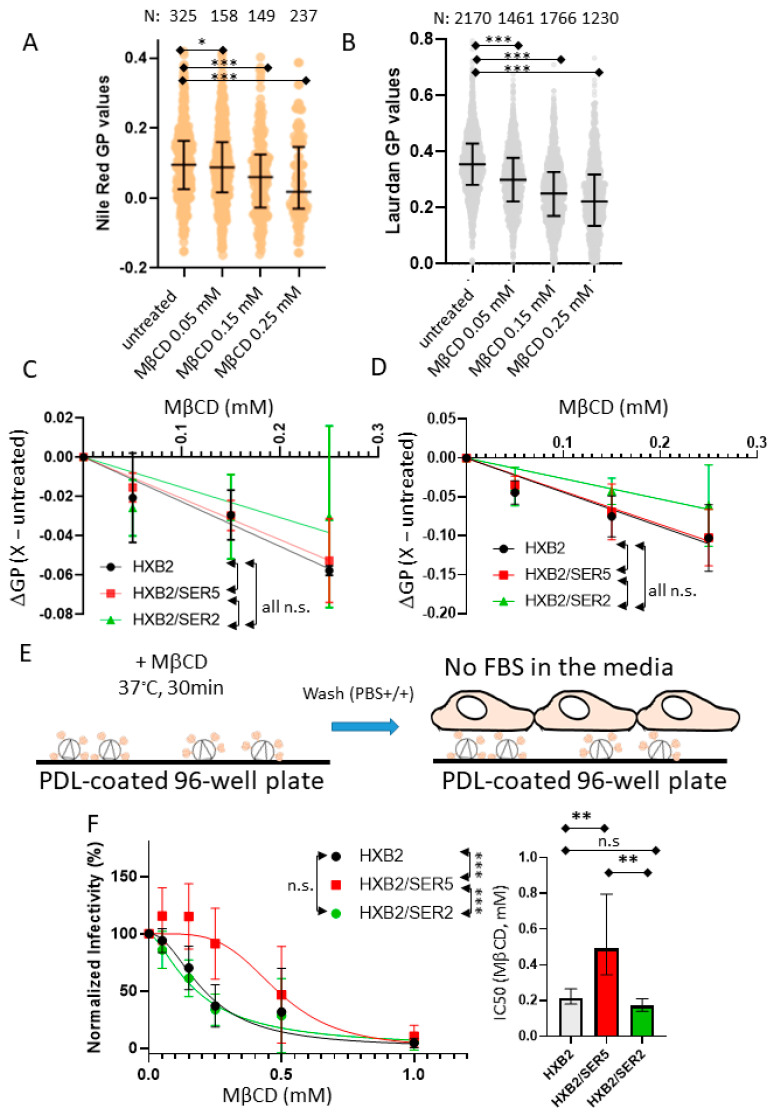
Effects of cholesterol extraction on lipid order and infectivity of control and SERINC containing pseudoviruses. Lipid order measurements of MβCD-treated viruses stained with Nile Red (**A**,**C**) and Laurdan (**B**,**D**). Examples of Nile Red (**A**) and Laurdan (**B**) staining on attached viruses. Horizontal black lines on the scatterplot represent 1st, 2nd, and 3rd quantiles. N, the number of viruses analyzed per experiment, is shown above the graphs. Statistics analysis was conducted using Kolmogorov–Smirnov test. (**C**,**D**) Means and SD of ∆GP values of 3 independent measurements. Each ΔGP was calculated by subtracting the median of the untreated viruses from the median of the 0.05-0.25 mM MβCD-treated viruses, ΔGP = X − untreated. Each virus was compared by 2-way ANOVA repeated measures. (**E**) Experimental workflow of the infectivity inhibition assay with cholesterol depletion by methyl-β-cyclodextrin (MβCD). After viruses were attached and treated with MβCD, TZM-bl cells were overlayed on the viruses after PBS+/+ wash. (**F**) (Left) Inhibition curves of control-, SER5-, and SER2-incorporated viruses treated with MβCD. At least 3 independent virus preparations were treated with MβCD for the infectivity inhibition assay and compared by 2-way ANOVA repeated measures. Infectivity data were normalized by untreated viruses and fitted using the Hill–Langmuir equation. (Right) IC_50_ values were derived using the fitting with an estimated 95% confidence interval within the top and bottom whiskers and compared using student’s *t*-test. n.s., *p* > 0.05; *, 0.05 > *p* > 0.01; **, 0.01 > *p* > 0.001 ***, *p* < 0.001.

**Figure 4 viruses-14-01636-f004:**
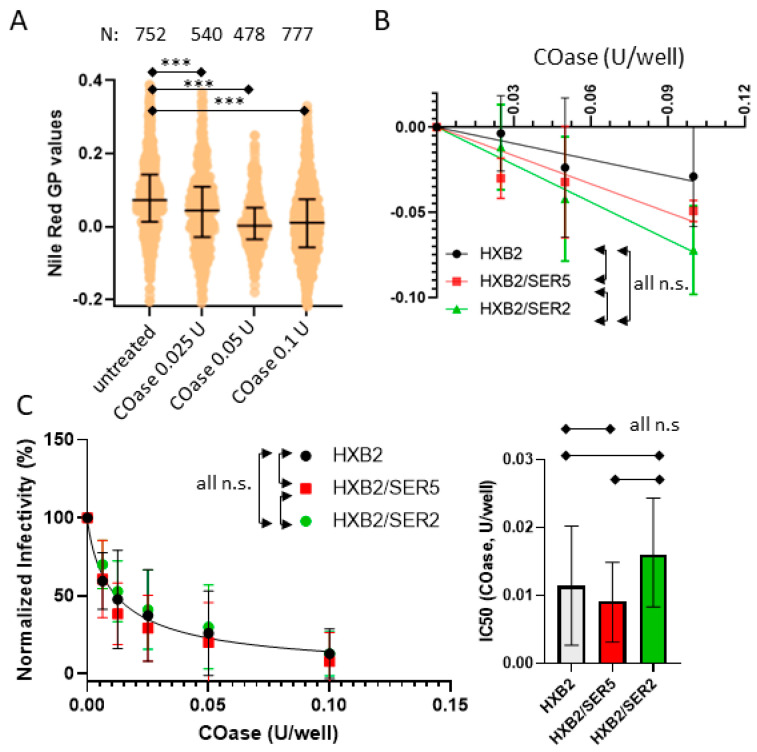
Effects of cholesterol oxidation on lipid order and infectivity of pseudoviruses. (**A**) Lipid order measurement with Nile Red staining and infectivity measurement of cholesterol oxidase (COase)-treated viruses stained with Nile Red. Horizontal black lines on the scatterplot represent 1st, 2nd, and 3rd quantiles. N, the number of viruses analyzed per experiment, is shown above the graphs. Statistical analysis was conducted using the Kolmogorov–Smirnov test. (**B**) Means and SD (error bars) of ΔGP between 3 independent measurements. Each ΔGP was calculated by subtracting the median of the untreated viruses from the median of the 0.025–0.1 U/well Coase-treated viruses, ΔGP = X-untreated. Each virus was compared by 2-way ANOVA repeated measures. (**C**) (Left) Inhibition curves of control-, SER5-, and SER2-incorporated viruses treated with COase. At least 3 independent virus preparations were treated with COase for the infectivity inhibition assay and compared by 2-way ANOVA repeated measures. Infectivity data were normalized by untreated viruses and fitted using the Hill–Langmuir equation. (Right) IC_50_ values were derived using the fitting with an estimated 95% confidence interval within the top and bottom whiskers and compared using student’s *t*-test. n.s., *p* > 0.05; ***, *p* < 0.001.

**Figure 5 viruses-14-01636-f005:**
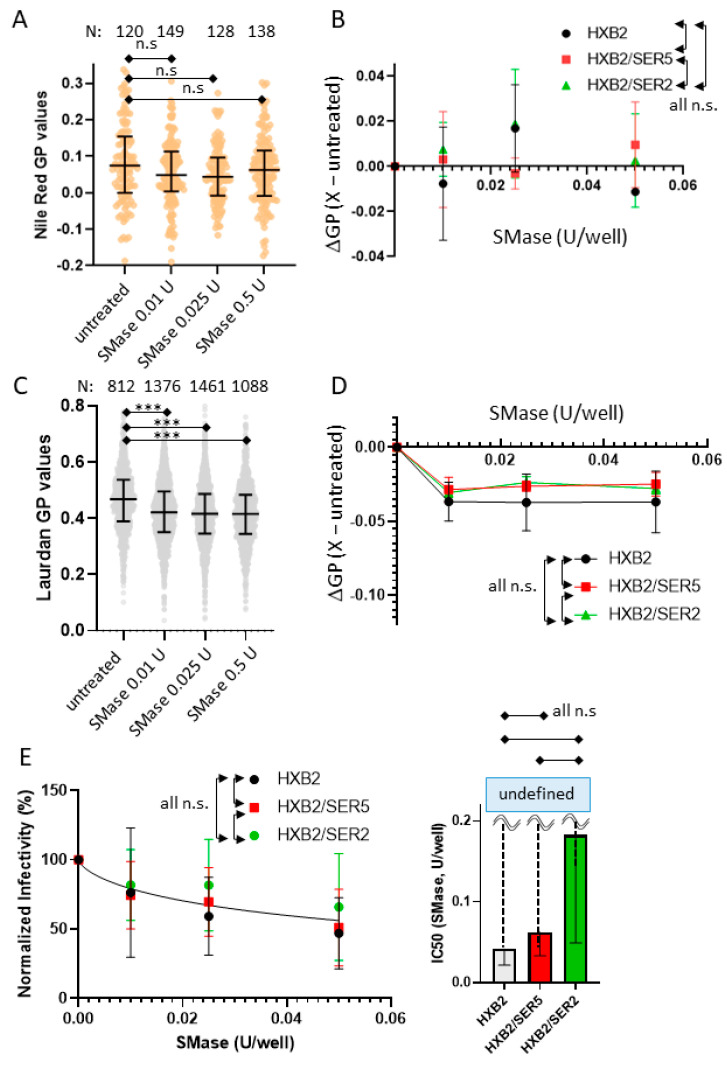
Assessing the effects of sphingomyelinase (SMase) treatment on virus lipid order and infectivity. (**A**) Nile Red staining of attached viruses (**B**) Means and standard deviations of ΔGP from 2 independent Nile Red measurements. Each virus was compared by 2-way ANOVA repeated measures. (**C**) Laurdan staining of attached viruses. Statistics analysis was conducted using Kolmogorov–Smirnov test. (**D**) Means and standard deviations of ΔGP from 2 independent Laurdan measurements. From (**B**,**D**), each ΔGP was calculated by subtracting the median of the untreated viruses from the median of the 0.01–0.05 U/well SMase-treated viruses, ΔGP = X-untreated. Each virus was compared by 2-way ANOVA repeated measures. For both (**A**,**C**), horizontal black lines on the scatterplot represent 1st, 2nd, and 3rd quantiles. N, the number of viruses analyzed per experiment, is shown above the graphs. (**E**) (Left) Inhibition of infectivity of WT-, SER5-, and SER2-incorporated viruses treated with SMase. At least 3 independent virus preparations were treated with SMase for the infectivity inhibition assay and compared by 2-way ANOVA repeated measures. Infectivity data were normalized by untreated viruses and fitted using the Hill–Langmuir equation. (Right) IC_50_ values were derived using the fitting with an estimated 95% confidence interval within the top and bottom whiskers and compared using student’s *t*-test. n.s., *p* > 0.05; ***, *p* <0.001.

**Figure 6 viruses-14-01636-f006:**
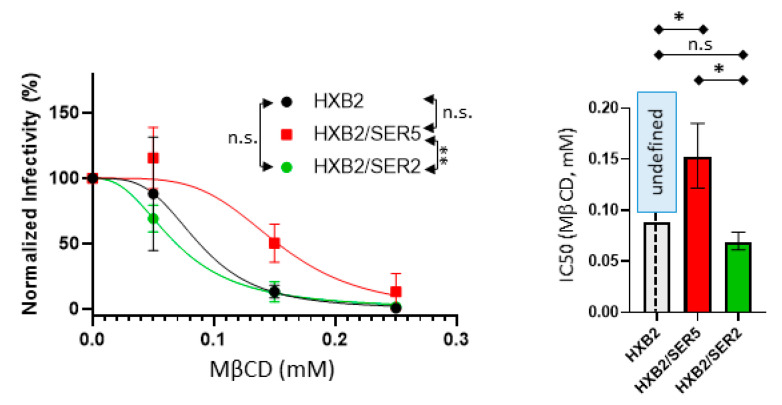
Inhibition of infectivity assays first treated (0.05 U/well) using SMase and then MβCD. (Left) 3 independent virus preparations were sequentially treated with SMase and MβCD for the infectivity inhibition assay and compared by 2-way ANOVA repeated measures. Infectivity data were normalized by SMase treatment only and fitted using the Hill–Langmuir equation. (Right) IC50 values were derived using the fitting with an estimated 95% confidence interval within the top and bottom whiskers and compared using student’s *t*-test. n.s., *p* > 0.05; *, 0.05 > *p* > 0.01; **, 0.01 > *p* > 0.001.

**Figure 7 viruses-14-01636-f007:**
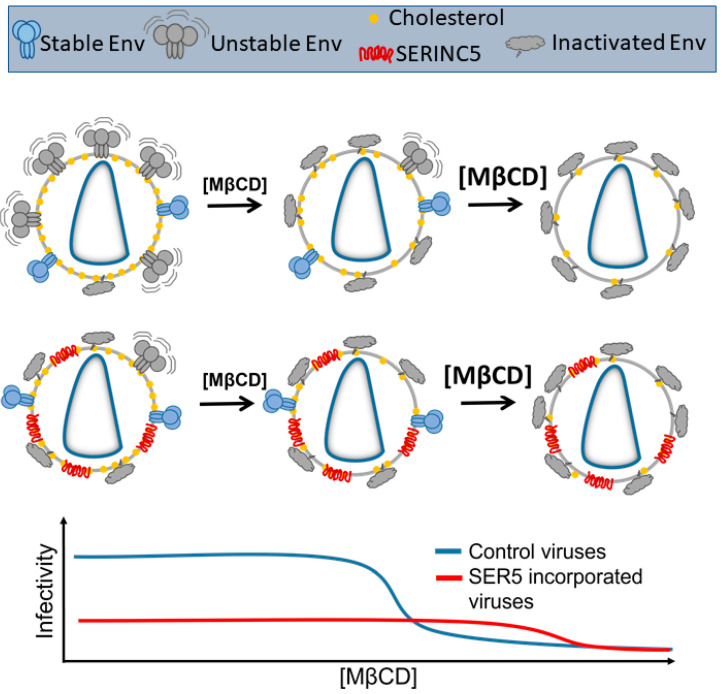
Proposed model for SER5-mediated restriction of HIV-1 Envs. We propose that HIV-1 pseudoviruses are decorated with both stable (represented by blue) and unstable (represented by grey) subpopulations of Envs. SER5 can preferentially inactivate the unstable subpopulation with minimal effect on the stable Envs. Stable Envs are likely more resistant to cholesterol extraction, resulting in the apparent resistance to MβCD observed in SER5 viruses.

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
