# Peer review of "SERINC5-Mediated Restriction of HIV-1 Infectivity Correlates with Resistance to Cholesterol Extraction but Not with Lipid Order of Viral Membrane"

_viruses, 2022, doi:10.3390/v14081636_

Round 1
Reviewer 1 Report
Raghunath et al studied the infectivity effect of cholesterol extraction to HIV-1 particles containing the restriction factor SER5. In addition, they study whether other lipid content is affected by SER5 in the particle. All results except for MBCD(cholesterol extraction) experiment did not show an effect.
The resistance to cholesterol extraction performed with MBCD results in a mild phenotype (maybe 2-fold on the IC50, and this graph has a very big standard deviation for HXB2/SER5, see Fig3F ); therefore, the phenotype is not clear (Figure 3F). This is a difficult experiment since viruses with SER5 are very restricted (dynamic range of the infection is very small), so it is not clear what are the real levels of infection(not normalized) that led the author to plot the graph for Figure 3F. In addition, the graph should consider the SER5-independent inhibition of infection by MBCD. It would be better to use a less restricted HIV-1 envelope by SER5, so the infectivity range increases.
Major
Figure 2 needs to show expression of SER5 in cells and viral particles, so we can understand the levels at which no change in lipids is observed. Preferable one would like to have equal levels of expression or a SER5 mutant that does not restrict.
Figure 3 again needs expression of SER5 and SER2 in particles, and in producing cells. The effect could be due to different amounts of SER proteins in the viral particle.
Figure 3, real infections used to plot Fig3F should be presented.
Figure 4 needs expression levels of SER2 and 5 in particles.
Minor
Line 327, it seems there is an error that precludes the understanding of this paragraph.
Explain the modified luciferase assay in the main manuscript.
Author Response
Reviewer #1: (Reviewer comments underlined)
- Raghunath et al studied the infectivity effect of cholesterol extraction to HIV-1 particles containing the restriction factor SER5. In addition, they study whether other lipid content is affected by SER5 in the particle. All results except for MBCD (cholesterol extraction) experiment did not show an effect. The resistance to cholesterol extraction performed with MBCD results in a mild phenotype (maybe 2-fold on the IC50, and this graph has a very big standard deviation for HXB2/SER5, see Fig3F); therefore, the phenotype is not clear (Figure 3F). –
Response: We appreciate the reviewer’s attention to detail. While we agree with the reviewer that the standard deviation for SER5 viruses in Figure 3F are relatively large (due to variations in viral preparations), we would like to emphasize that the difference in IC50 is highly statistically significant (P<0.01). This statistical significance was found to be true with multiple independent biological replicates. Hence, we are confident in our claim that there is a clear phenotype, with SER5 viruses exhibiting a decreased sensitivity to cholesterol extraction, in comparison to SER2 and control viruses. As a more general point, one would not expect large differences in resistance to cholesterol extraction, which ultimately leads to major changes in membrane structure and properties. Therefore, a 2-fold increase in MbCD IC50 appears fairly strong.
- This is a difficult experiment since viruses with SER5 are very restricted (dynamic range of the infection is very small), so it is not clear what are the real levels of infection (not normalized) that led the author to plot the graph for Figure 3F. –
Response: We thank the reviewer for acknowledging the difficulty with performing these experiments. We chose to display normalized data for the reader to better appreciate the effect of MbCD titration on the infectivity of the samples. For the sake of clarity, our updated manuscript includes non-normalized luminescence data of the different virus preparations in the Supplemental Information document (Figure S8A). We have also included a representative figure describing raw, non-normalized SER5 infectivity changes with MbCD titration, relative to background (Figure S8B).
We have also included another figure (Figure S9) to show non-normalized luminescence data for control, SER5 and SER2 viruses so that the reader can better appreciate the changes to luminescence levels with increasing [MbCD]. As shown in the figures, care was taken to ensure that the signal levels from SER5 viruses were always maintained at levels above the background (equal to or greater than 10-fold relative to background for most of the experiments, barring [MbCD] >=0.5mM). Regardless of signal normalization, it is obvious that SER5 containing viruses exhibit a clear resistance to cholesterol extraction by MbCD.
We have also included several lines in the main text dedicated to addressing the dynamic range of our assay and can be found in section 3.4.
- In addition, the graph should consider the SER5-independent inhibition of infection by MBCD. It would be better to use a less restricted HIV-1 envelope by SER5, so the infectivity range increases. –
Response: We have already considered SER5-independent inhibition of infection by MbCD. This was the primary rationale for using SER2 incorporated and control viruses. As mentioned in the main text, SER2 is a homologous protein belonging to the SERINC family of proteins. For all our lipid manipulation experiments, we compared SER5 and SER2 across [MbCD]. We also added additional S.I. figures to demonstrate that the luminescence signals of SER5 containing viruses were ≥10-fold above background. We are not sure if infectivity measurements on SER-resistant viruses would add any value to our study, as SER5-resistant Envs are also likely to be inherently less sensitive to cholesterol depletion [Salimi et al; PMID 31757809].
- Figure 2 needs to show expression of SER5 in cells and viral particles, so we can understand the levels at which no change in lipids is observed. –
Response: The reviewer raises a valid concern about expression levels of SER5 in pseudovirus samples. While our previous iteration of the S.I. figures included immunofluorescence data indicating the same, we have decided to revise the supplemental figures for clarity (Figures S5 and S6). As shown in the figures, our particles consistently express SER5 and SER2, albeit with some variation between independent preparations. The expression levels were comparable to previous studies from our group and others, who have shown effective HIV-1 restriction at such concentrations (Sood et. Al; PMID: 28179429 & Chen et al; PMID: 32441921), with SER2 incorporation consistently exceeding that of SER5. We have also included several lines in the main text to address the expression levels of SER5 and SER2 that can be found in section 3.4.
We are not sure if measuring the expression levels of SER5 in producer cells will add any value to our study, which is focused exclusively on fully formed and infective pseudoviruses.
- Preferable one would like to have equal levels of expression or a SER5 mutant that does not restrict –
Response: As mentioned above, SER2 is homologous to SER5 and incorporates at somewhat higher levels than SER5. We believe it fulfils a similar purpose to the SER5 mutant that the reviewer is referring to.
- Figure 3 again needs expression of SER5 and SER2 in particles, and in producing cells. The effect could be due to different amounts of SER proteins in the viral particle. –
Response: We have addressed this critique in the revised version of the supplemental figures (Figures S5 and S6). We have also addressed this directly in the main text (section 3.4). If anything, SER2 incorporation into virions is somewhat greater than SER5, highlighting the specific effect of SER5 on resistance to viral lipid manipulations.
- Figure 3, real infections used to plot Fig3F should be presented. -
Response: We have addressed this critique by including non-normalized luminescence data in the revised S.I. document (Figures S8 and S9).
- Figure 4 needs expression levels of SER2 and 5 in particles. -
Response: We have addressed this critique in the revised version of the supplemental figures (Figures S6 and S7). We have also addressed this directly in the main text (section 3.4).
- Line 327, it seems there is an error that precludes the understanding of this paragraph. –
Response: We thank the reviewer for pointing this error out. This was a result of re-formatting from article submission. We have rectified the mistake in the updated version of the main text.
- Explain the modified luciferase assay in the main manuscript. -
Response: We have made modifications to the main text to explain our assay with better clarity. We have removed the word “modified” from our text, as we found that to be misleading. We thank the reviewer for feedback in this regard.
Reviewer 2 Report
The manuscript by Raghunath and colleagues tested the possibility that the antiviral protein SERINC5 affects lipid order of HIV-1 and/or cholesterol. Lipid order was assessed using the two probes Laurdan and Nile Red. Effects of cholesterol was tested using cholesterol depletion or cholesterol oxidation. The authors find that lipid order is not affected by SERINC5. In contrast, SERINC5 incorporation makes viruses less sensitive to cholesterol depletion. Based on these findings, the authors propose that a subpopulation of more stable HIV-1 envelope glycoproteins residing on SERINC5 containing virus particles are more resistant to cholesterol depletion.
The work was performed rigorously, the tools were first validated, controls such as comparing mature an immature virus particles included to connect and reproduce previous literature. It is convincingly demonstrated that SERINC5 doesn’t’ significantly change lipid order. In contrast, the authors demonstrate that SERINC5 containing HIV-1 becomes more resistant to cholesterol depletion. This is the main finding of the paper. Oxidation of cholesterol didn’t have an effect. Hydrolyzing sphingomyelin to ceramid also didn’t affect the action of SERINC5. However, the sensitivity of HIV-1 to cholesterol depletion was increased following sphingomyelin hydrolysis.
This is study is well done, of interest to the readership of Viruses, and I have only two minor comments, first, line 327: There is text missing after the 3.1 subtitle, and second, while the literature review on the role of cholesterol in HIV-1 infectivity in the introduction is outstanding, the classic Brugger lipodomics study (Ref. 13) compared virus lipids to total cellular lipids (all mixed including nucleus and ER), not to plasma membrane lipids where HIV-1 viruses bud. If HIV-1 lipids are compared to plasma membrane lipids and released exosomes, the main lipids being enriched are phosphoinositides (PMIDs: 18799574, 31776383). The Brugger lab also updated this in their more recent 31776383 paper.
Author Response
Reviewer #2: (Reviewer comments underlined)
- The manuscript by Raghunath and colleagues tested the possibility that the antiviral protein SERINC5 affects lipid order of HIV-1 and/or cholesterol. Lipid order was assessed using the two probes Laurdan and Nile Red. Effects of cholesterol was tested using cholesterol depletion or cholesterol oxidation. The authors find that lipid order is not affected by SERINC5. In contrast, SERINC5 incorporation makes viruses less sensitive to cholesterol depletion. Based on these findings, the authors propose that a subpopulation of more stable HIV-1 envelope glycoproteins residing on SERINC5 containing virus particles are more resistant to cholesterol depletion.
The work was performed rigorously, the tools were first validated, controls such as comparing mature an immature virus particles included to connect and reproduce previous literature. It is convincingly demonstrated that SERINC5 doesn’t’ significantly change lipid order. In contrast, the authors demonstrate that SERINC5 containing HIV-1 becomes more resistant to cholesterol depletion. This is the main finding of the paper. Oxidation of cholesterol didn’t have an effect. Hydrolyzing sphingomyelin to ceramid also didn’t affect the action of SERINC5. However, the sensitivity of HIV-1 to cholesterol depletion was increased following sphingomyelin hydrolysis.
This is study is well done, of interest to the readership of Viruses, and I have only two minor comments, -
Response: We thank the reviewer for his/her kind words, and an accurate description of the work that was presented in the paper.
- first, line 327: There is text missing after the 3.1 subtitle, -
Response: We thank the reviewer for pointing this error out. We believe this was caused due to formatting issues during article submission. We have rectified the mistake in the updated version of the main text.
- and second, while the literature review on the role of cholesterol in HIV-1 infectivity in the introduction is outstanding, the classic Brugger lipodomics study (Ref. 13) compared virus lipids to total cellular lipids (all mixed including nucleus and ER), not to plasma membrane lipids where HIV-1 viruses bud. If HIV-1 lipids are compared to plasma membrane lipids and released exosomes, the main lipids being enriched are phosphoinositides (PMIDs: 18799574, 31776383). The Brugger lab also updated this in their more recent 31776383 paper. -
Response: We thank the reviewer for their excellent suggestions with regards to the literature to be cited. We agree with the comment, and we have revised the manuscript to directly reference the work and include citations for the both the papers suggested by the reviewer.
Reviewer 3 Report
SERINC5 is a host restriction factor that strongly blocks HIV-1 entry by attacking the Env trimers, but the mechanism is still unclear. In addition, the SERINC5 cellular function remains unclear, which makes its antiviral mechanism even more promiscuous. In this manuscript, the Melikyan lab investigated whether SERINC5 changes the lipid order of cellular and viral membrane using SERINC3 as a control and concluded that it does not have such activity. However, they found that SERINC5 could make viral membrane become more resistant to cholesterol extraction, which is interesting. Overall, this study is well designed, and the manuscript is well written, which provides some new insights into SERINC5 antiviral mechanism and its cellular function. I only have a few minor concerns that must be addressed.
1. Line 44-48, please include another reference (PMID: 11516650) in the introduction, which reports HIV-1 budding from lipid rafts.
2. Line 550, where is Fig.5F?
3. Line 680, please fix the problem.
4. Line 683, why Ref #37 is cited here? Please include the correct reference (PMID: 31043528).
Author Response
We thank the reviewer for the kind comments, and for providing an accurate description of our work.
- Line 44-48, please include another reference (PMID: 11516650) in the introduction, which reports HIV-1 budding from lipid rafts. –
Our response:
We thank the reviewer for the suggestion regarding additional citation. We have cited the article suggested by the reviewer in the updated version of our manuscript.
- Line 550, where is Fig.5F? –
Our response:
We thank the reviewer for pointing out the error in our manuscript. Figure 5F was used to call attention to the inset of Figure 5E. We have removed our references to 5F in the updated version of the manuscript.
- Line 680, please fix the problem. –
Our response:
The updated version of our manuscript has the updated citation for the article that the reviewer is referring to. At the time of submitting this manuscript, the article by Kirschman et al was still under review and has since been published.
- Line 683, why Ref #37 is cited here? Please include the correct reference (PMID: 31043528) -
Our response:
We thank the reviewer for catching this error. The updated version of our manuscript now carries the correct reference as suggested by the reviewer.
Round 2
Reviewer 1 Report
Unfortunately the only positive phenotype of this story is a 2-fold phenotype with large standard deviations, which is the result of extracting cholesterol of already very restricted particles(very small dynamic range, change of infectivity)(Figure S8B). This should be mention and discussed in the results section. This is a known problem in the SER5 field.
minor but important
The excessive use of normalization clouds the ability of the reader to understand results and significance.
Author Response
- Unfortunately the only positive phenotype of this story is a 2-fold phenotype with large standard deviations, which is the result of extracting cholesterol of already very restricted particles(very small dynamic range, change of infectivity)(Figure S8B). This should be mention and discussed in the results section. This is a known problem in the SER5 field. –
We respectfully disagree with the reviewer’s suggestion that the roughly 2-fold phenotype that we observe is not a significant result. We also disagree with the notion that our infectivity assay has a small dynamic range. We show clearly, in our supplemental data figures, that the large error bars are not just a consequence of the small dynamic range of our infectivity assay. Despite the relatively large error bars due to variable results seen in independent viral preparations, it is abundantly clear from multiple biological repetitions that we observe a statistically significant increase in IC50 for SER5 containing virions for MBCD treatment. This is a very unexpected discovery that sheds new light on possible SER5 function.
From the raw data (shown in Figure S8, in the previous version of our manuscript), even with a diminished signal, the samples containing SER5 viruses exhibit luminescence intensity values that are clearly greater than the background levels (>= 10-fold for samples treated with 0.25mM or less of MBCD). Hence, we do not understand the reviewer’s comment that our data is representative of a systemic issue with the “SER5 field”.
- The excessive use of normalization clouds the ability of the reader to understand results and significance. -
We have added a sentence clearly highlighting that the data in the main manuscript is normalized.
The revised version of the manuscript now directly references the non-normalized data in Figure S8 and S9 in response to the previous suggestion from reviewer #1.